# Tryptophanyl-Transfer RNA Synthetase Is Involved in a Negative Feedback Loop Mitigating Interferon-γ-Induced Gene Expression

**DOI:** 10.3390/cells13020180

**Published:** 2024-01-17

**Authors:** Ikrame Lazar, Ido Livneh, Aaron Ciechanover, Bertrand Fabre

**Affiliations:** 1The Rappaport Technion Integrated Cancer Center (R-TICC) and the Rappaport Faculty of Medicine and Research Institute, Technion-Israel Institute of Technology, Haifa 3109601, Israel; ikrame.lazar@univ-tlse3.fr (I.L.); idoliv@campus.technion.ac.il (I.L.); aaroncie@technion.ac.il (A.C.); 2MCD, Centre de Biologie Intégrative (CBI), CNRS, UT3, Université de Toulouse, 31400 Toulouse, France; 3Laboratoire de Recherche en Sciences Végétales (LRSV), CNRS/UT3/INPT, 31320 Auzeville-Tolosane, France

**Keywords:** tryptophanyl-tRNA synthetase, melanoma, interferon-γ, signal transducer and activator of transcription 1, proteomics, mass spectrometry

## Abstract

Aminoacyl-tRNA synthetases (aaRSs) are essential enzymes responsible for linking a transfer RNA (tRNA) with its cognate amino acid present in all the kingdoms of life. Besides their aminoacyl-tRNA synthetase activity, it was described that many of these enzymes can carry out non-canonical functions. They were shown to be involved in important biological processes such as metabolism, immunity, development, angiogenesis and tumorigenesis. In the present work, we provide evidence that tryptophanyl-tRNA synthetase might be involved in a negative feedback loop mitigating the expression of certain interferon-γ-induced genes. Mining the available TCGA and Gtex data, we found that *WARS* was highly expressed in cutaneous melanoma (SKCM) compared to other cancers and is of good prognosis for this particular cancer type. *WARS* expression correlates with genes involved in antigen processing and presentation but also transcription factors involved in IFN-γ signaling such as *STAT1*. In addition, WARS was found in complex with STAT1 in A375 cells treated with IFN-γ. Finally, we showed that knocking down WARS expression during IFN-γ stimulation further increases the expression of *GBP2*, *APOL1*, *ISG15*, *HLA-A* and *IDO1*.

## 1. Introduction

Cutaneous melanoma is an aggressive form of skin cancer with a high propensity to metastasize and colonize distant organs [1]. Recent developments in immunotherapy for the treatment of advanced and recurring melanoma have led to a striking improvement in patient outcomes [2]. Antibodies directed at programmed cell death protein 1 receptor (PD1) and its ligand PDL-1 as well as cytotoxic T-lymphocyte-associated protein 4 (CTLA-4) strongly activate the immune system, leading to tumor recognition and destruction [2]. However, certain patients are resistant to such treatments [3]. Recent studies point toward defects in the interferon-γ (IFN-γ) signaling as a potential cause of resistance to immunotherapy [3,4]. IFN-γ is a key pro-inflammatory cytokine that plays an important role in modulating the immune response [5]. Cellular responses to IFN-γ stimulation are mediated by the IFN-γ receptor (IFN-γR), which activates downstream intracellular signaling cascades, ultimately leading to the expression of several genes involved in the immune response [5]. This regulation of gene expression is mediated through an array of transcription factors, such as Signal transducer and activator of transcription 1 (STAT1), STAT3, STAT5, IRF1 (Interferon regulatory factor 1) and NF-κB (Nuclear factor kappa-light-chain-enhancer of activated B cells) [5]. IFN-γ signaling is regulated by a negative feedback loop involving the suppressor of cytokine signaling 1 (SOCS1) [6]. The link between interferon-γ and melanoma is still not fully understood [7]. Indeed, IFN-γ signaling is frequently down-regulated in melanoma and seems important for the response to anti-CTLA4 and anti-PD1 therapies [3,4] but might play a dual role in melanoma or other types of cancers (pro- or anti-tumorigenic) [4,7,8,9,10,11,12]. In addition, mutations of several genes involved in the response to IFN-γ in tumor cells render them resistant to certain therapies (e.g., anti-CTLA-4 therapy) [12]. Clinical trials using IFN-γ as a therapeutic agent in melanoma were performed in the late 1980s and early 1990s, with mixed results (little to no response from the patients to the therapy) [13,14,15]. Recently, a combination of IFN-γ and nivolumab (monoclonal antibody against PD-1) showed modest benefits with one patient achieving a complete response and five patients having stable disease [16].

Aminoacyl-tRNA synthetases (aaRSs) are essential enzymes responsible for linking a transfer RNA (tRNA) with its cognate amino acid and are present in all the kingdoms of life [17]. Given their important function, genetic-variation-disrupting aaRS activities are often associated with diseases [17,18]. Interestingly, it was shown that, besides their aminoacyl-tRNA synthetase activity, several aaRSs can carry non-canonical functions [19]. They were shown to be involved in important biological processes such as metabolism, immunity, development, angiogenesis and tumorigenesis [17,19]. As an example, the tryptophanyl-tRNA synthetase (also known as WARS, WRS or TrpRS) was described as having several functions besides its aminoacyl-tRNA synthetase activity. Indeed, it was shown to act as an angiogenesis antagonist by inhibiting VE-cadherin [20,21], to regulate the cellular tryptophan influx [22,23,24], to promote the polyADP-ribosylation of DNA-PKcs by PARP1 [25] or to be secreted in response to the infection by pathogens [26]. WARS is known to be up-regulated upon IFN-γ stimulation in many human cell types [25]. Here, we further investigated the role of WARS following IFN-γ stimulation. We notably showed that WARS becomes the 11th-most abundant protein in the A375 melanoma cell line treated with IFN-γ. We also observed, using TCGA data, that a high expression of *WARS* in skin cutaneous melanoma (SKCM) tumors is indicative of a better survival of patients. Finally, our results demonstrate an interaction between WARS and STAT1 in IFN-γ-treated cells and an as-yet-unidentified role of WARS as a negative regulator of the expression of a set of IFN-γ-induced genes.

## 2. Materials and Methods

### 2.1. Cell Culture

A375 cells were obtained from ATCC and grown in DMEM at 37 °C in a humidified atmosphere with 5% CO_2_ as described in [27].

### 2.2. Plasmids

shRNA-targeting WARS or WARS-Flag cloning was performed using NEBuilder^®^ HiFi DNA Assembly (New England Biolabs, Ipswich, UK) according to the manufacturer’s instructions. shRNA-targeting WARS (CCAGCACCTACCAGTAATCAT) was cloned into pLKO lentiviral vector containing a tetracycline-induced promoter, also encoding for the Tet-Responsive element as well as blasticidin resistance. WARS coding sequence was amplified using cDNA generated from A375 cell line (forward: AACGGCCGCCAGTGTGCTGGATGCCCAACAGTGAGCCCGCATC; reverse: TGACACTATAGAATAGGGCC TTACTTATCATCATCATCCTTATAGTCCTGAAAGTCGAAGGACAGCT) and cloned into pcDNA3.1 expression vector.

### 2.3. Sample Preparation and Mass Spectrometry (MS) for Proteome Analysis

Cell lysis, sample preparation and mass spectrometry analysis were performed as described previously [28].

### 2.4. Proteome Data Analysis

The raw files were analyzed using MaxQuant [29] version 1.6.0.1 software with the settings and database similar to [28]. The results were imported into Microsoft Excel for further analysis. Protein group intensities were normalized by their median intensities. Differentially expressed proteins were selected based on a ratio threshold of two-fold and a *p*-value from Welch’s *t*-test of 0.05. Proteins with intensity values in at least three replicates in one condition and at least one intensity value superior to the Q1 calculated based on all intensity values, but no intensity value in all the replicates of the other condition, were also considered as being differentially regulated. Protein networks and GO term enrichment analysis were performed with STRING v11 [30].

### 2.5. Sample Preparation and Mass Spectrometry (MS) for Interactome Analysis

A375 cells were transfected for 36 h with pLKO or a plasmid coding for FLAG-WARS using CalFectin (SignaGen Laboratories, Frederick, USA) and treated with 100 ng/mL of interferon-γ for 48 h, washed twice with phosphate-buffered saline and frozen at −80 °C. Cells were lysed in lysis buffer (50 mM Tris-HCl pH 8, 150 mM NaCl, 0.5% Nonidet P-40 and 1× Protease Inhibitor Cocktail (Sigma Aldrich, Burlington, MA, USA)) and incubated for 15 min at 4 °C. The lysate was then centrifuged for 10 min at 14,000× *g*. The supernatant was incubated overnight at 4 °C with beads conjugated with FLAG antibody (Sigma Aldrich, Burlington, MA, USA). Beads were pelleted by centrifugation, washed six times with lysis buffer and incubated with 3× Flag Peptide (150 mg/mL, Sigma Aldrich, Burlington, MA, USA) twice for 1 h at 4 °C. Finally, proteins in the eluates were subjected to tryptic digestion and analyzed by LC/MS-MS, as described previously [31].

### 2.6. Interactome Analysis

The raw files were analyzed using MaxQuant [29] version 1.6.0.1 software with the settings and database similar to [28]. The results were imported into Microsoft Excel for further analysis. In order to select WARS’s interactors, we used the following strategy. Ratios were calculated between the minimal intensity value measured among the FLAG-WARS immunoprecipitation (IP) replicates and the maximal intensity value measured among the control IP replicates. Such ratio was calculated for each protein for which an intensity value was observed in at least two replicates. Proteins with ratios of at least 2 were considered enriched in the IP WARS. Proteins with intensity values in at least three replicates in the WARS IP and at least one intensity value superior to the Q1 calculated based on all intensity values, but no intensity value in all the replicates of the control IP, were also considered as being WARS-interacting proteins. Protein networks and GO term enrichment analysis were performed with STRING v11 [30].

### 2.7. Co-Immunoprecipitation of Endogenous WARS and STAT1

A375 cells were cross-linked with 0.1% formaldehyde as previously described [32]. Cells were lysed in lysis buffer (50 mM Tris-HCl pH 8, 150 mM NaCl, 0.5% Nonidet P-40 and 1× Protease Inhibitor Cocktail (Sigma Aldrich, Burlington, MA, USA)) and incubated for 15 min at 4 °C. The lysate was then centrifuged for 10 min at 14,000× *g*. The supernatant was incubated overnight at 4 °C with beads (Pierce™ Crosslink IP Kit, Thermo Fisher Scientific, Waltham, MA, USA) conjugated with cross-linked anti-WARS (Abcam, Cambridge, UK, ab92733), anti-STAT1 (Cell Signaling Technology, Danvers, MA, USA, 9172) or control (Cell Signaling Technology, 2729) antibodies. Beads were pelleted by centrifugation, washed six times with lysis buffer and eluted by boiling beads in 2× Laemmli buffer.

### 2.8. WARS Knock-Down and Interferon-γ Treatment

ShWARS plasmid and a non-targeting shControl vector were used to prepare viral particles, followed by A375 cell transduction and selection using 15 µg/mL Blasticidin for 7 days [33]. Cells were cultured in the presence of 500 ng/mL doxycycline for 5 days to induce WARS knock-down, later validated via Western blot. Cells were then treated with 100 ng/mL IFN-γ at the indicated periods (see figure legends), or with different concentrations of IFN-γ for 48 h in dose–response experiments. Cells were rinsed with phosphate-buffered saline (PBS) and stored at −80 °C until further processing.

### 2.9. Western Blot Analysis

Frozen cells were lysed in a sodium dodecyl sulfate (SDS)-based buffer [34], sonicated and boiled for 5 min at 95 °C. BCA assay (Thermo Fisher Scientific, Waltham, MA, USA) was used to measure protein concentration in each sample. 5× Laemmli buffer was added to 50 µg of proteins and the samples were boiled for 5 min at 95 °C before loading on an SDS-PAGE gel (acrylamide concentration of 4% for the stacking gel and 12% for the resolving gel). After migration, proteins were transferred on a nitrocellulose membrane using the TransBlot^®^ Turbo™ Transfer System (Biorad, Hercules, CA, USA). The remaining free binding sites on the membrane were saturated with PBS containing 5% milk. Membranes were incubated overnight with anti-WARS (Abcam, Cambridge, UK, ab92733), anti-STAT1 (Cell Signaling Technology, Danvers, MA, USA, 9172), anti-PhosphoSTAT1 (Tyr701) (Cell Signaling Technology, Danvers, MA, USA, 9167), anti-IDO1 (Cell Signaling Technology, Danvers, MA, USA, 86630) or anti-GAPDH (Abcam, Cambridge, UK, ab9484) primary antibodies (dilution advised by the vendor). Membranes were rinsed 4 times with PBS containing 0.1% tween (PBS/T) and incubated with secondary antibodies against mouse (Santa Cruz Biotechnology, Dallas, TX, USA, sc-2005) or rabbit (Santa Cruz, sc-2004) for 1 h. Membranes were then rinsed 4 times with PBS/T and revelation was performed using ECL (Biorad, Hercules, CA, USA). Densitometric quantification was performed using the ImageJ software (latest v.1.5.4) [35].

### 2.10. Reverse Transcription-Quantitative Polymerase Chain Reaction

RNA extraction, cDNA synthesis and qPCR were performed as described previously [22]. Primer sequences are provided in Appendix A. Primer efficiencies were checked before performing the experiments.

### 2.11. Statistical Analysis

All analyses for the Western blots and reverse transcription-quantitative polymerase chain reaction were performed using Microsoft Excel. Values are represented as means ± standard deviation. The statistical significance of differences between the means was evaluated using Student’s *t* tests. *p* values below 0.05 (*) and *p* < 0.01 (**) were considered significant.

## 3. Results

### 3.1. Interferon-γ Reshapes the Proteome of A375 Cells and Strongly Increases Tryptophanyl-tRNA Synthetase (WARS) Abundance

To the best of our knowledge, no study has been conducted to monitor changes induced by IFN-γ in melanoma cells at the proteome level. Thus, in order to evaluate how IFN-γ impacts the proteome of melanoma cells, A375 cells were treated with 100 nM of IFN-γ for 48 h and the modulation of protein expression was analyzed using mass spectrometry (MS)-based proteomics (Figure 1A). Quantitative data were obtained for 3889 proteins with 253 proteins significantly changing in abundance by at least 2-fold between the IFN-γ and control condition (Figure 1B and Appendix A). Among the 76 proteins up-regulated, enrichments in genes involved in the “Immune system process”, “Antigen processing and presentation”, “Immune response” or “Response to interferon-gamma” were found (Figure 1C and Appendix A) and were expected as they were known to be induced by IFN-γ from previous transcriptomics data from other melanoma cell lines [8]. When focusing on proteins down-regulated by IFN-γ, we found proteins involved in the cell cycle and cell division (Figure 1D and Appendix A). It was indeed shown that IFN-γ limits the proliferation of melanoma cells [36], which we also observed in A375 cells (Appendix A).

One striking result was the induction of the tryptophanyl-tRNA synthetase (WARS) which was up-regulated 28 times (Figure 2A). Although it was previously observed in several cell types that the WARS level is increased following IFN-γ treatment [25], here, using the intensity-based absolute quantification (iBAQ) as an approximation of the absolute abundance of a protein [37,38], we show that WARS becomes the 11th-most abundant protein in A375 following 48 h of IFN-γ treatment (Figure 2B). As described previously, WARS is a moonlighting protein with several described functions [39]. We were thus interested in further characterizing the function of WARS associated with IFN-γ, especially given the high expression level of this protein following stimulation with this cytokine.

### 3.2. WARS Is Highly Expressed in Cutaneous Melanoma (SKCM) and Its Expression Level Correlates with Genes Involved in the Immune Response and with Improved Patient Survival

We used GEPIA (Gene Expression Profiling Interactive Analysis, version 1) [40] to mine the available TCGA and Gtex data regarding WARS expression in cancer and normal cells (the abbreviations are detailed for each tumor type in Appendix A).

Strikingly, WARS was highly expressed in skin cutaneous melanoma (SKCM) compared to other cancers (Figure 3A) or to normal tissue (Figure 3B). We then looked at the survival of SKCM patients with a high or low expression of *WARS*. As displayed in Figure 3C, SKCM patients with a high expression of *WARS* have a better chance of survival compared to patients with a low expression of *WARS* (*p* = 0.00014, hazard ratio = 0.59).

We then used GEPIA to identify the top-50 genes correlating with *WARS* (Appendix A) in the TCGA and Gtex data. Surprisingly, most of these genes are involved in the immune response (Figure 4A). Among the genes correlating with WARS were genes involved in antigen processing and presentation (*PSMB8*, *PSMB9 PSME1*, *PSME2*, *TAP1*, *TAP2*, *HLA-B*, *HLA-C*, *HLA-E* and *HLA-F*) but also transcription factors such as STAT1, IRF1 and NLRC5 (Figure 4A and Appendix A). The gene correlating the most with *WARS* expression was *STAT1* (Spearman correlation coefficient = 0.74) (Figure 4B). STAT1 is known to be the key transcription activator of *WARS* during IFN-γ stimulation [41] but might also be important for *WARS* expression under basal conditions.

Next, we assessed the survival of patients expressing high or low levels of each of these 50 genes for each cancer type (Appendix A). Strikingly, most of these genes are of good prognosis for SKCM (Appendix A). This might highlight that high levels of IFN-γ in SKCM might be beneficial for patient survival. However, a high expression of this gene set seems to be of bad prognosis for two other cancers, low-grade glioma (LGG) and uveal melanoma (UVM) (Appendix A), in which too much IFN-γ might be deleterious for the survival of patients. The improved survival in SKCM patients with a high *WARS* expression (Figure 3C) is probably a consequence of a high level of IFN-γ in the tumor of these patients (which induced high expression of *WARS*), which seems of good prognosis in this particular type of cancer.

### 3.3. The Definition of WARS Interactome Highlights an Interaction between WARS and the Transcription Factor STAT1

In order to characterize the possible alternative function of WARS, other than its t-RNA synthetase activity, we defined its interactome. A375 cells were either transfected with a plasmid expressing WARS tagged with Flag or an empty vector (Figure 5A). The cells were then treated for 48 h with 100 nM of IFN-γ. Anti-Flag immunoprecipitation was conducted on protein extracts followed by elution with a Flag peptide, and the immunoprecipitated proteins were digested with trypsin and analyzed by mass spectrometry (Figure 5A). A total of 158 proteins were found interacting with WARS (Figure 5B and Appendix A), highlighting probable moonlighting activities of WARS. Among them were proteins involved in “tRNA aminoacylation of protein translation”, “Response to unfolded protein”, “Mitochondrial transport”, “Tryptophan transport” and “Immune effector process” (Figure 5B). WARS was notably found in complex with other t-RNA synthetases, namely Phenylalanyl (FARSA), Glycyl (GARS) and Aspartyl (DARS) -tRNA synthetase (Figure 5B). The interaction with the latter was previously observed in MCF7 cells (human breast cancer cell line) [42]. DARS is a member of the multi-enzyme aminoacyl-tRNA synthetase complex (MSC) [43], which is not the case for WARS. These data suggest that certain t-RNA synthetases might interact with each other outside of the MSC. In addition, the heteromeric amino-acid transporter LAT1–CD98hc (or SLC7A5 and SLC3A2), which is known to mediate the import of tryptophan from the extracellular medium [44], was found interacting with WARS (Figure 5B). This is particularly interesting as it was recently shown that WARS is important for tryptophan uptake in conditions where this amino acid is depleted from cells, such as IFN-γ stimulation [22,23]. However, the transporter associated with this tryptophan influx was not clearly defined [23]. Our data suggest that the interaction between WARS and LAT1–CD98hc might increase the tryptophan uptake into human cells.

When combining the interactome, expression correlation data and IFN-γ up-regulated proteins (proteome), two proteins emerged (excluding WARS from this analysis): the cytosolic aminopeptidase LAP3 and the transcription factor STAT1 (Figure 5C). We were particularly interested in a potential interaction between WARS and STAT1 as the latter is one of the key transcription factors involved in the IFN-γ response [45]. In addition, as shown previously, STAT1 is the most correlated gene to WARS in the TCGA and Gtex datasets (Figure 4B). In order to validate the interaction observed by MS, we performed co-immunoprecipitation experiments of WARS and STAT1 followed by Western blot analysis. When the endogenous WARS was immunoprecipitated from A375 cells treated with IFN-γ for 48 h, we could observe an interaction with the endogenous STAT1 (Figure 5D). The reverse co-immunoprecipitation (endogenous STAT1 immunoprecipitated from A375 cells treated with IFN-γ for 48 h) confirmed the interaction between endogenous STAT1 and WARS (Figure 5E).

### 3.4. The Knock-Down of WARS Further Increases the Expression of a Set of Interferon-γ-Induced Genes

The interferon-γ signaling is a tightly regulated process, and prolonged IFNγ-induced signaling in newborn mice is lethal [46]. An example of a negative feedback loop of IFN-γ signaling is the expression of suppressor of cytokine signaling 1 (SOCS1). Following prolonged stimulation by the cytokine, SOCS1 binds to JAK1 and JAK2 (Janus Kinase1 and 2) promoting their inhibition and their degradation by the proteasome, and finally, the inhibition of IFN-γ signaling [47]. Given that several mechanisms exist to regulate IFN-γ-induced signaling and the associated immune response (e.g., PD1/PD-L1 interaction), and the fact that WARS is found in complex with STAT1 in A375 cells, we decided to investigate a possible role of WARS in IFN-γ-induced signaling. We generated A375 cells transduced with a plasmid coding for doxycycline-inducible shRNA-targeting WARS (shWARS). The induction of this shRNA strongly decreases WARS expression at the RNA (Appendix A) and protein level (Appendix A). We first monitored the cell viability of A375 cells transduced with pLKO or shWARS upon IFN-γ treatment (Appendix A). We observed a slight (1.5-fold) improvement in cell viability in the presence of IFN-γ in the cells transduced with shWARS versus cells transduced with pLKO (Appendix A). Of note, cells transduced with shWARS also seem to grow slightly faster than cells transduced with pLKO when treated with the vehicle (Appendix A). We then monitored the effect of WARS knock-down on the expression level of genes known to be up-regulated by IFN-γ stimulation. Surprisingly, certain tested genes (*GBP2*, *ISG15 APOL1* and *HLA-A*) were further increased in A375 cells knocked-down for *WARS* and treated with IFN-γ (Figure 6A). This was particularly striking for indoleamine-pyrrole 2,3-dioxygenase (*IDO1*), which is involved in the kynurenine pathway and is a strong immunosuppressor [48], for which the expression increased by 290 times after 4 h of IFN-γ treatment in pLKO-transduced cells versus 687 times in WARS knock-down cells (Figure 6B). We then looked to see whether a similar trend was observed at the protein level for IDO1. A time course of 48 h was performed in cells transduced with pLKO or shRNA WARS and the level of IDO1 was monitored by Western blot. We did not observe any changes in STAT1 phosphorylation profile dependent on WARS knock-down (Figure 6C). However, the induction of IDO1 protein expression is stronger in WARS knock-down cells (Figure 6C and Appendix A). When performing an IFN-γ dose–response experiment, a higher protein expression level of IDO1 at 50 and 100 ng/mL of IFN-γ in WARS knock-down cells was observed (Figure 6D and Appendix A). With IDO1 being a key enzyme in the metabolization of tryptophan in kynurenine, its further up-regulation upon IFN-γ stimulation in cells with a low level of expression of WARS would fasten kynurenine production and thus increase immune system inhibition. Altogether, these data show that some IFN-γ-regulated genes are further induced in cells with a reduced expression level of WARS (Figure 6). Our data thus point toward a possible role of WARS in a negative feedback loop of the IFN-γ-induced signaling that might depend on its interaction with STAT1. It was previously shown in a wide variety of cell lines that, upon IFN-γ stimulation, WARS relocalizes in the nucleus [20]. This is also the case for STAT1 which is mainly cytosolic in basal conditions and is translocated into the nucleus when cells are stimulated with IFN-γ [41]. This change in localization of WARS was associated with a role of WARS in the polyADP-ribosylation of DNA-PKcs by PARP1 [25]. However, given the very high abundance of WARS observed in A375 treated with IFN-γ (Figure 2), one can imagine that a fraction of WARS might interact with STAT1 in the nucleus in IFN-γ-stimulated cells. Nevertheless, further experiments would be needed to precisely determine the molecular mechanisms involved.

## 4. Conclusions

In the present work, we provide evidence that tryptophanyl-tRNA synthetase might be involved in a negative feedback loop mitigating the expression of certain interferon-γ-induced genes. Notably, an as-yet-unidentified interaction between WARS and STAT1 was discovered, suggesting a function of WARS associated with this key transcription factor (Figure 5). In addition, we show that knocking down WARS expression during IFN-γ stimulation further increases the expression of the interferon-γ-induced genes *GBP2*, *ISG15*, *HLA-A* and *IDO1* (Figure 6A,B). In the case of *IDO1*, this trend was also observed at the protein level (Figure 6C,D). These data point toward a role of WARS as a negative regulator of the interferon-γ pathway. Whether WARS can negatively modulate the transcriptional activity of STAT1 to restrain the expression of IFN-γ-regulated genes still needs to be further investigated. It also remains to be determined if WARS t-RNA synthetase activity or other domains of this protein are important in this process. All in all, WARS might represent a new potential future target to modulate the immune response and more particularly to alleviate its suppression by cancers such as melanoma [39].

## Figures and Tables

**Figure 1 cells-13-00180-f001:**
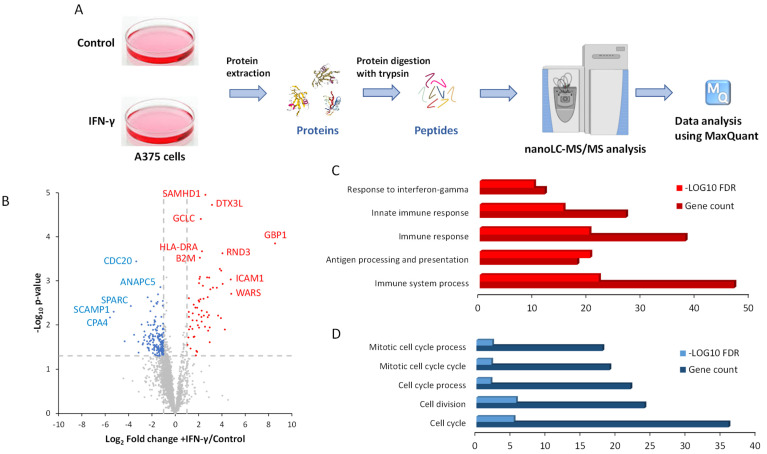
Label-free quantitative proteomics workflow to identify modulation in the proteome of A375 cells following interferon-γ treatment. (**A**) A375 cells treated with 100 ng/mL of IFN-γ or treated with the vehicle solution (control) (n = 4 for each condition) for 48 h were lysed, and proteins were extracted, precipitated and digested with trypsin. Peptides were run in data-dependent acquisition mode on a Q Exactive HF-X mass spectrometer. The resulting raw data were analyzed with MaxQuant. (**B**) Volcano plot representing the log2 ratio (+IFN-γ/control) for each protein quantified and the corresponding ^−^Log10 *p*-value. The blue, red and gray dots represent the proteins more abundant in control cells, more abundant in IFN-γ-treated cells or not differentially expressed, respectively. The grey dashed lines represent the fold change and *p*-value thresholds. Some of the names of the most differentially regulated proteins are displayed on the volcano plot. (**C**,**D**) Results from Gene Ontology pathway analysis for the proteins more abundant in IFN-γ-treated cells (**C**) and more abundant in control cells (**D**).

**Figure 2 cells-13-00180-f002:**
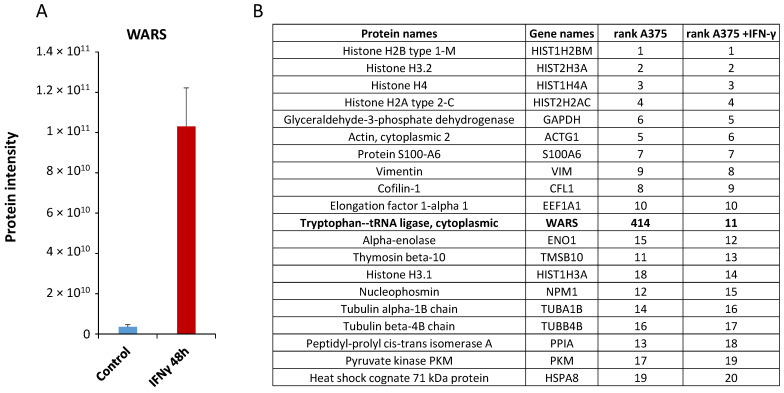
WARS is highly up-regulated in A375 cells treated with IFN-γ. (**A**) Histogram displaying the measured protein intensity for WARS in control or IFN-γ-treated cells (errors bars represent standard deviations). (**B**) Table showing the top-20 most abundant proteins (based on their iBAQ values) in A375 cells treated with IFN-γ and their corresponding ranking in control A375 cells.

**Figure 3 cells-13-00180-f003:**
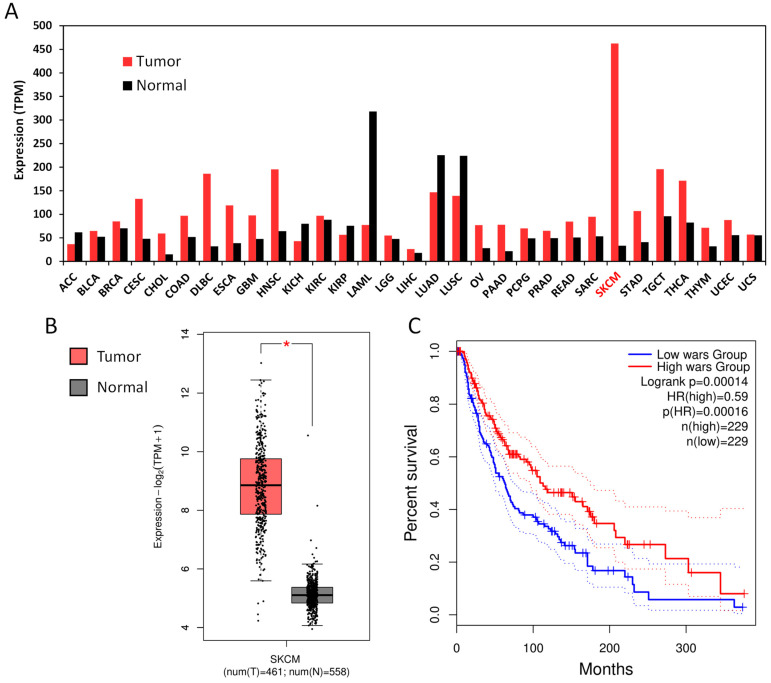
WARS is highly expressed in skin cutaneous melanoma and patients with high expression of WARS display better survival. (**A**) *WARS* expression in different cancers and corresponding normal tissues (based on TCGA and Gtex data). The abbreviations are detailed for each tumor type in Appendix A. SKCM is highlighted in red. (**B**) Expression levels of *WARS* in SKCM tumor versus normal tissue. The asterisk represents a significant difference at a *p*-value < 0.05. (**C**) Survival analysis of patients expressing high and low levels of *WARS*.

**Figure 4 cells-13-00180-f004:**
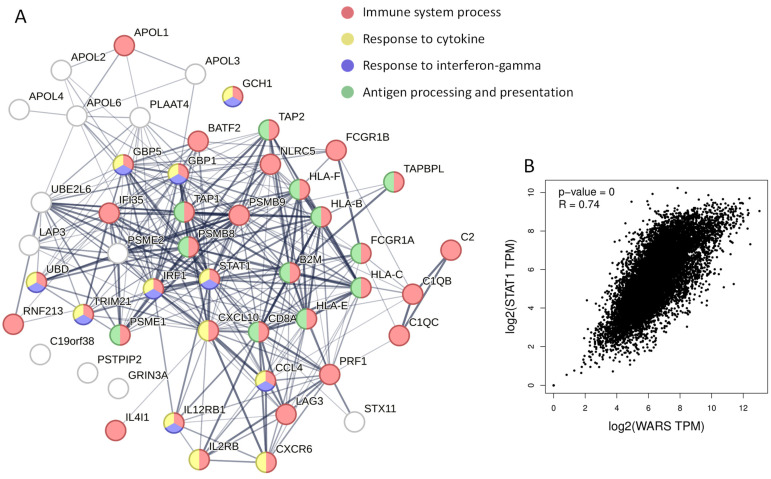
*WARS* correlates with genes involved in the immune response. (**A**) STRING network of the top-50 genes (excluding *WARS*) correlating with WARS from an analysis using GEPIA [40] (TCGA and Gtex data). (**B**) Gene expression correlation between *WARS* and *STAT1*. The R corresponds to the Spearman ranked correlation coefficient.

**Figure 5 cells-13-00180-f005:**
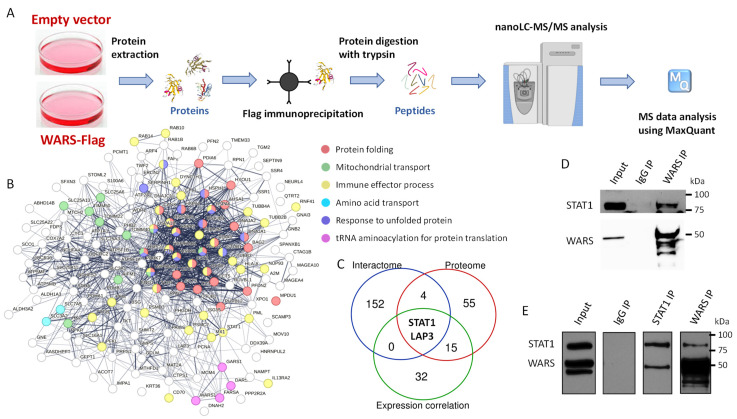
The interactome of WARS reveals an interaction with the transcription factor STAT1. (**A**) A375 cells were transfected with pLKO or a plasmid coding for WARS tagged with a Flag and treated with 100 ng/mL of IFN-γ or treated with the vehicle solution (control) (n = for each condition) for 48 h. The cells were lysed and WARS Flag was immunoprecipitated using an anti-Flag antibody. Beads were incubated with Flag peptides, eluates were digested with trypsin and peptides were run in data-dependent acquisition mode on a Q Exactive HF-X mass spectrometer. The resulting raw data were analyzed with MaxQuant. (**B**) STRING network of the proteins identified as interacting with WARS. The different colors represent proteins from different biological processes. (**C**) Venn diagram displaying the overlap between the genes correlating with WARS from the GEPIA analysis, the proteins up-regulated by IFN-γ in the whole proteomics experiment and WARS-associated proteins from the interactome. (**D**) Immunoprecipitation of endogenous WARS in IFN-γ-treated A375 cells followed by WARS and STAT1 Western blots. (**E**) Immunoprecipitation of endogenous WARS and STAT1 in IFN-γ-treated A375 cells and WARS and STAT1 Western blots.

**Figure 6 cells-13-00180-f006:**
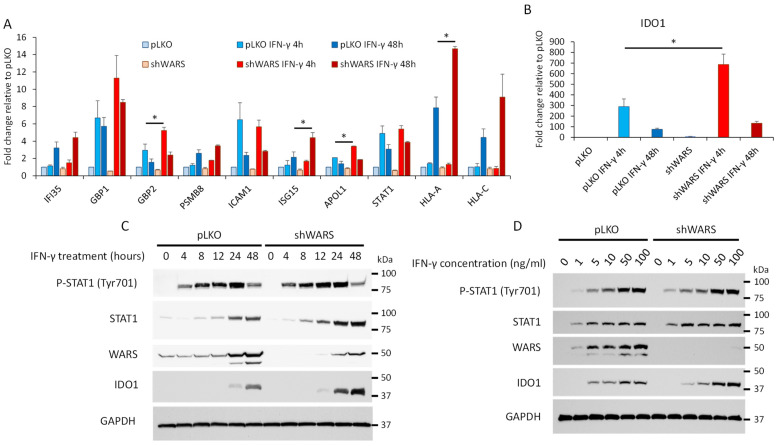
Knocking down WARS further increases the expression of a set of interferon-γ-induced genes. (**A**,**B**) A375 cells were transfected with pLKO or a plasmid coding for a shRNA-targeting WARS and treated with 100 ng/mL of IFN-γ or treated with the vehicle solution (control) for 4 or 48 h. RNAs were extracted and a reverse transcription-quantitative polymerase chain reaction was performed for several genes known to be induced by IFN-γ. Histograms showing the changes in expression of these genes (**A**), and IDO1 (**B**), are displayed (errors bars represent standard deviations). The asterisks represent a significant difference at a *p*-value < 0.05. (**C**) Time course experiment of A375 cells transfected with pLKO or a plasmid coding for a shRNA-targeting WARS and treated with 100 ng/mL of IFN-γ or treated with the vehicle solution. Proteins were extracted and a Western blot analysis was performed to monitor Phospho-STAT1, STAT1, WARS, IDO1 and GAPDH (loading control). (**D**) The same as in (**C**) but with a dose–response (0 to 100 ng/mL of IFN-γ for 48 h) performed instead of a time course.

## Data Availability

All the mass spectrometry data have been deposited in the MassIVE repository with the dataset identifier MSV000093688.

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
