# Peer review of "Tryptophanyl-Transfer RNA Synthetase Is Involved in a Negative Feedback Loop Mitigating Interferon-γ-Induced Gene Expression"

_cells, 2024, doi:10.3390/cells13020180_

Round 1

Reviewer 1 Report

Comments and Suggestions for Authors

Lazar et al., report the strong upregulation of Tryptophanyl-tRNA synthetase (WARS) in skin cutaneous melanoma cells following treatment with IFNgamma. Interestingly, WARS expression increases from being the 414th highest expressed protein to the 11th highest one within 48 hours. Comparison with publicly available cancer expression data sets support the relevance of this correlation during the pathogenesis of skin cancer as WARS levels are elevated. To understand the function of this drastic increase, the authors analyzed the interactome of WARS and found that it associates with an amino acid importer (LAT1–CD98hc) and a transcription factor (STAT1). STAT1 is one of the major regulators in the cellular response to IFNgamma and surprisingly, WARS1 knockdown decreased the increase in expression of IFNgamma sensitive genes.

The main findings are very interesting, and the reported changes are convincing. The following points should be addressed to improve data presentation:

1.      Figure 1B would benefit from labeling at least some of the points. Otherwise, it just shows that the obtained data can be visualized in a volcano plot. Lines to indicate the cut-off values would be beneficial. It would make the manuscript more accessible if the red and blue genes would be indicated as such in Supplementary Table S2 to spare the reader the task to repeat the analysis steps.

2.      Figure 3: Tumor type abbreviations should be listed in the manuscript or at least SKCM highlighted.

3.      Figure 4 is very messy and hard to understand – I would consider moving it to the Supplementary information. At the very least, individual proteins should be highlighted in Figure 4A. Figure 4C could be reduced to SKCM if it is supposed to be the main focus of the paper or again, SKCM could be at least highlighted.

4.      As a note, I think the first impression of the quality of the paper as a whole suffers from the lack of curation and analysis given to publicly available data sets. I think the mass spec results and western blots that the authors provide are very clear and of good quality but at the moment, the reader has to work their way through difficult to assess large datasets in suboptimal presentation to appreciate them. I acknowledge their importance for the narrative of the report but they should be presented with more care.

5.      Figure 5B: None of the labels can be read. What is the purpose of this figure? Why is the medium in Figure 5A blue?

6.      Figure 5D: It looks as if bands are cut off in the STAT1 blot. I am assuming that they are IgG bands and they should be shown and labeled as such to confirm that equal amounts of antibodies were used.

7.      Figure S3: Does WARS knock down affect cell viability upon IFNgamma treatment?

8.      Molecular weight should be labeled in all western blots.

9.      Conclusions drawn from the quantitative assessment of Figure 6C and D need to be supported by densitometric analysis of western blots, ideally in replicates.

10.   The finding that WARS, FARSA, and GARS, and DARS interact is quite interesting, especially as DARS is a member of the multisynthetase complex. Is there support for this finding in the literature? Would the conditions under which the IP was performed support indirect interactions?

11.   Given that exposure time is likely not comparable between S6B and C I am sure what the point is of including S6C – the authors show clearly in Figure 6C and D that the WARS response to IFN is dampened by shWARS. Figure S6B should show a larger area as the lower band is cut off in the IFN treated samples.

12.   Statistical analysis is missing from qRT-PCR results. Generally, a section explaining statistical methods should be added.

13.   Generally, Figure legends are very limited and could be expanded on to provide more detail. 

Comments on the Quality of English Language

The quality of English Language is good, some copy editing might help avoid smaller errors in grammar and typos.

Reviewer 2 Report

Comments and Suggestions for Authors

In the manuscript entitled „Tryptophanyl-tRNA synthetase is involved in a negative feed-back loop mitigating interferon-γ induced gene expression“, the authors conducted a study to investigate the role of tryptophanyl-tRNA synthetase (WARS) after stimulation of A375 melanoma cells with IFN-γ. In addition, WARS expression in different cancer types was analysed using the Gene Expression Profiling Interactive Analysis database and a significant correlation between Wars and the transcription factor Stat1 was found. Finally, the authors have shown that knocking-down WARS in A375 cells increases the expression of a number of IFN-γ-induced genes and for IDO1 this was confirmed at the protein level.

It may be beneficial for the manuscript to have the data on the use of IFN-γ in melanoma therapy in the Introduction section.

The results presented in this study lack a more detailed discussion, e.g.  what would be the implications of increased expression of IDO1 in WARS knock-down A375 cells treated with IFN-γ, or what is the putative mechanism underpinning the better survival of patients with higher WARS expression?

I would recommend formulating the conclusions (Section Conclusions) more clearly in order to emphasize the novelty of the study results and underline their impact.

Properly introduce abbreviations in the manuscript, that is, full term first and then abbreviations in parenthesis. Also, some abbreviations are earlier in the text, and full terms are later e.g. abbreviation SOCS1 line 42 and full term is in line 299.

Line 64 abbreviation SKCM should be introduced
